# Rape Straw Biochar Application Enhances Cadmium Immobilization by Promoting Formation of Sulfide and Poorly Crystallized Fe Oxide in Paddy Soils

**Rui Yuan** [1,2]**, Tianren Si** [1,2]**, Qingquan Lu** [1,2]**, Cheng Liu** [1,2]**, Rongjun Bian** [1,2]**, Xiaoyu Liu** [1,2]**, Xuhui Zhang** [1,2]**, Jufeng Zheng** [1,2]**, Kun Cheng** [1,2]**, Stephen Joseph** [1,2,3]**, Yan Wang** [1,2]**, Lianqing Li** [1,2,*] **and Genxing Pan** [1,2]

[1] Institute of Resources, Ecosystem and Environment of Agriculture, Center of Biochar and Green Agriculture, Nanjing Agricultural University, 1 Weigang, Nanjing 210095, China; 2019203037@njau.edu.cn (R.Y.)

[2] Jiangsu Collaborative Innovation Center for Solid Organic Waste Resource Utilization, Nanjing 210095, China

[3] School of Materials Science and Engineering, University of New South Wales, Sydney, NSW 2052, Australia

[*] Correspondence: lqli@njau.edu.cn; Tel./Fax: +86-25-8439-8657

**Abstract:** The mechanisms of rape straw biochar that affect the fixation of cadmium (Cd) in paddy soil by influencing redox of iron and sulfur are unclear. Several anaerobic incubation experiments were carried out using Cd-contaminated paddy soils (LY and ZZ). Rape straw biochar at pyrolysis temperatures of 450 °C (LRSB) and 800 °C (HRSB) was selected as the soil remediation agent. The electron exchange capacity and electrical conductivity were higher in HRSB than those in LRSB. The lower pe + pH in HRSB enhanced Fe oxide reduction, with a maximum increase in $Fe^{2+}$ of 46.0% in ZZ. Compared to treatment without biochar (CK), the poorly crystallized Fe oxide ($Fe_o$) in HRSB increased by 16.8% in ZZ. This induced Cd bound to Fe, and Mn oxides fraction (Fe-Mn Cd) increased by 42.5%. The $SO_4^{2-}$-S content in LRSB was 4.6 times that of HRSB. LRSB addition increased acid-volatile sulfide by 46.4% and 48.9% in LY and ZZ soils, respectively, compared to CK. This resulted in an increase in sulfide's contribution to Cd fixation, with values rising from 24.2% to 37.8% in LY and 19.1% to 29.8% in ZZ. Overall, LRSB reduced Cd mobility by forming more sulfide, while HRSB increased Fe-Mn Cd by increasing $Fe_o$.

**Keywords:** rape straw biochar; sulfide; Fe-Mn oxides Cd; $Fe_o/Fe_d$

## 1. Introduction

Cadmium (Cd) is classified as a heavy metal with high biological toxicity in agricultural systems [1]. Crops grown in Cd-contaminated soils have the potential to accumulate Cd in edible portions [2]. Hence, the consumption of rice containing elevated levels of Cd in the diet poses a significant risk to human health [3].

The amount of Cd in rice is influenced by the Cd mobility in soils [4]. The rice cultivation is distinguished by intermittent flooding, which exerts an influence on the speciation of heavy metals in soils [5]. Heterotrophic bacteria are primarily responsible for the reductions in iron during flooding conditions; $Fe^{3+}$ is reduced to $Fe^{2+}$ by electron transfer in microorganisms [6,7]. Previous research had highlighted the significance of sulfate reduction in reducing the Cd solubility. Cd K-edge X-ray absorption spectroscopy revealed that Cd precipitated as barely soluble cadmium sulfides [8]. Furthermore, the $Fe^{2+}$ in the solution reacted with sulfide to produce a secondary form of iron sulfide (FeS), such as mackinawite [9], which has been found to reduce the concentration of dissolved Cd by precipitation [10]. Iron oxides are primarily found in the environment as minerals such as ferrihydrite, goethite, and hematite. The hydroxylation of dissolved $Fe^{2+}$ in soil solutions resulted in the formation of iron hydroxides, oxyhydroxides or oxides [11,12]. The addition of dissolved $Fe^{2+}$ could react with the ferrihydrite to further form lepidocrocite [13]. The presence of a high specific surface area and an abundance of functional groups (such as

hydroxyl groups) on the surface of iron oxides facilitates the availability of numerous sorption sites for the binding of Cd [14]. Thus, precipitation/reduction/hydroxylation of iron oxides and the formation of sulfide are connected to the Cd bioavailability in paddy soils [15,16].

Biochar is considered to be a promising amendment to Cd-polluted soils due to its larger specific surface area, microporous structure [17,18], and active functional groups [19]. Furthermore, the feedstock type of biochar is a crucial determinant in affecting the Cd passivation in soil. Biochar derived from pig carcass exhibited a total phosphorus content of 8.3%, which effectively reduced Cd mobility through the formation of insoluble Cd-phosphate precipitates [20]. Rice straw biochar was regarded as Si-rich biochar ($SiO_2$ of 16.65%), which could stabilize Cd in soil through co-precipitation [21]. Rape straw had a significantly higher sulfate content compared to other crop straws [22]. It was reported that biochar produced from oilseed rape straws showed a total sulfur content of 0.31%, surpassing the sulfur content found in biochar derived from corn stalk (0.08%) and rice straw (0.26%) [23–25]. In addition, sulfate availability in biochar decreased with increasing pyrolysis temperature [26]. The water-soluble sulfate in corn straw biochar at 300 °C was higher than those pyrolyzed at 500 °C and 700 °C [27].

Biochar has been found to undergo reversible electron transfer with both Fe oxide and reducing bacteria, acting as an electron acceptor and donor in the process [28,29]. The redox characteristics of biochar and the ensuing redox-mediated interactions between biochar and Fe in soils are contingent upon factors such as feedstock and pyrolysis conditions [30]. Pinewood biochar pyrolyzed at 400 °C presented higher capacity in donating electrons than that pyrolyzed at 500 °C or 600 °C [31]. Rape straw biochar application was shown to have higher influence on enhancing microbial Fe reduction than that in rice straw biochar application [24]. Application of rice straw biochar led to an augmentation of poorly crystallized Fe oxide within the coarse particulate organic matter [32]. After amending in soil for three years, Cd was observed bonding with the Fe mineral phases inside the contaminated biochar particle [33]. However, the mechanism of biochar action in the process of Cd immobilization by the iron and sulfur cycle under flooded soil conditions is rarely studied.

The objective of the present study was to investigate the mechanism of Cd immobilization in flooded paddy soils by rape straw biochar (RSB) at different pyrolysis temperatures. We hypothesized that RSB at different pyrolysis temperatures could promote Cd immobilization by forming more poorly crystallized Fe oxide or sulfide.

## 2. Materials and Methods

### 2.1. Soil and Biochar

Paddy soils classified as red soil were collected from Cd-contaminated paddy fields located in Liuyang (LY), and Zhuzhou (ZZ), Hunan Province from the plow layer (0–20 cm). The soils were air-dried and then sieved through a 1 mm sieve. Biochar was produced by pyrolyzing rape straw at 450 °C (LRSB) and 800 °C (HRSB) for 2 h under oxygen-limited conditions at a rate of 5 °C min$^{-1}$. After cooling for 3 h, the biochar was ground to a fine powder and sieved to a diameter of less than 1 mm before being used in the batch. Tables 1 and 2 present the properties of the soils and biochar.

**Table 1.** Properties of soils.

| Properties | pH | SOC (g kg$^{-1}$) | DOC (g kg$^{-1}$) | CEC (cmol kg$^{-1}$) | Total Cd (mg kg$^{-1}$) | $SO_4^{2-}$-S (mg kg$^{-1}$) | Fe$_o$ (g kg$^{-1}$) | Fe$_d$ (g kg$^{-1}$) | Sand (%) | Loam (%) | Clay (%) |
|---|---|---|---|---|---|---|---|---|---|---|---|
| LY | 5.80 | 13.4 | 0.14 | 10.4 | 0.545 | 42.6 | 8.8 | 41.4 | 22.4 | 51.3 | 27.5 |
| ZZ | 5.49 | 14.1 | 0.19 | 11.2 | 0.813 | 43.6 | 4.3 | 30.4 | 16.2 | 61.5 | 23.1 |

Data are the mean ± SD (*n* = 3) and expressed on a dry weight (dw) basis.

**Table 2.** Properties of biochar.

| Properties | pH | SOC (g kg$^{-1}$) | DOC (g kg$^{-1}$) | CEC (cmol kg$^{-1}$) | Total Cd (mg kg$^{-1}$) | SO$_4^{2-}$-S (g kg$^{-1}$) | EEC (mmol e$^{-1}$(g $_{biochar}$)$^{-1}$) | EC (scm$^{-1}$) |
|---|---|---|---|---|---|---|---|---|
| LRSB | 8.2 | 588 | 3.1 | 50.4 | 0.513 | 7.4 | 1.11 | n.d |
| HRSB | 9.2 | 635 | 1.0 | 44.2 | 0.504 | 1.6 | 2.37 | 1.39 |

Data are the mean $\pm$ SD ($n$ = 3) and expressed on a dry weight (dw) basis.

### 2.2. Anaerobic Incubation

One hundred fifty grams of soil was placed in a serum bottle (250 mL, diameter: 5 cm). The 2% ($w/w$) rape straw biochar (LRSB, HRSB) was added and uniformly mixed. The soil without amendments was CK. The 150 mL Milli-Q water gave a 1:1 ($w/v$) soil/water mixture and formed a 5 cm layer of standing water above the soil surface. The bottles were purged with N$_2$ for 30 min to remove oxygen from the headspace before being sealed tightly and incubated in an incubator (25 $\pm$ 1 °C). Each treatment set repetitions three times. Pore water was collected on the 1, 3, 7, 10, 20, 30, and 45th days of flooding incubation. A soil solution sampler was inserted from the top to the center of the soil layer. Ten milliliters of pore water were sampled, and the environmental variables (Cd$^{2+}$, SO$_4^{2-}$, Fe$^{2+}$) were evaluated. Redox potential (Eh) and pH were tested in soils. The soil samples on the 45th day were taken to measure the soil-free Fe oxide, poorly crystallized Fe oxide, and Cd fractions of BCR.

The purpose of the second incubation experiment was to measure the sulfide contribution to Cd immobilization. Soil (150 g) with a 2% RSB mixture in 250 mL Milli-Q water was mixed. The addition of 20 mM molybdate was used to inhibit the activities of sulfate-reducing bacteria (+Mo) [7]. The soils were incubated under the flooded conditions as described above, and soil sampled on the 45th day. The concentrations of acid soluble Cd were measured.

### 2.3. Chemical Analyses

The determination of soil properties was carried out according to the procedure described by Lu [34]. Soil pH was measured in a suspension of 1:2.5 soil water ratio. The soil Eh was measured with an Eh meter and platinum electrodes (Thermo Scientific Orion, star A221, Waltham, MA, USA). Soil organic carbon (SOC) was determined using wet digestion by potassium dichromate. Soil cation exchange capacity (CEC) was extracted by 1 M ammonium acetate. The citrate-bicarbonate-dithionite(DCB) method was used to extract the free Fe oxide from the soil. The poorly crystallized Fe oxide was extracted using a 0.2 M ammonium oxalate solution. The HCl-extractable Fe$^{2+}$ and Fe$^{3+}$ were extracted by 0.5 M HCl solution [35]. The acid-volatile sulfide (AVS) in the soils was extracted with 10 mL of 6 M HCL. Then, the glass reactors were shaken (200 rpm) for 4 h, and solutions were quantified by the methylene blue method [36].

For determination of total Cd concentration, a portion (equivalent to 0.5 g) of each sample was digested with a mixed solution of HF-HNO$_3$-HClO$_4$ (10:2.5:2.5, $v$:$v$:$v$). The certified reference material (GBW07401) and blank samples were digested in each batch of digestions. The recovery of Cd was 102.6% throughout the analysis procedures.

The various fractions of Cd concentrations were detected with the methods of BCR [37]. The acid-soluble fraction included the Cd bound to carbonate and cation exchange site, which was obtained through soil extraction with 0.11 M acetic acid (acid-soluble Cd). The Cd bound to Fe and Mn oxides (Fe/Mn-Cd) was obtained by extracting with 0.5 M NH$_2$OH-HCl. The 30% ($m/v$) H$_2$O$_2$ was added into a soil and water bath for 1 h at 85 °C. Another 5 mL of 30% ($m/v$) H$_2$O$_2$ was used to digest the sample at 85 °C. Then, the soil was extracted with 1 M of ammonium acetate at pH 2 to test Cd in organic matter. The residual Cd was detected with HF-HNO$_3$-HClO$_4$ (10:2.5:2.5, $v$:$v$:$v$). The various fractions of Cd concentrations were measured by atomic absorption spectrometry (AAS). The total amount of Cd recovered from the soil was 82.1% through the BCR methods.

The pore water was acidified using concentrated HCl to 0.1% acid ($v/v$), and then a concentration of $SO_4^{2-}$ in pore water ($W\text{-}SO_4^{2-}$) was measured by turbidimetry. The pore water was acidified using concentrated HCl to 1% acid ($v/v$), and then a concentration of $Fe^{2+}$ in pore water ($W\text{-}Fe^{2+}$) was measured by HEPES ($C_8H_{18}N_2O_4S$) buffer containing 1 g $L^{-1}$ ferrozine; the concentration of $Cd^{2+}$ in pore water ($W\text{-}Cd^{2+}$) was measured by atomic absorption spectrometry (AAS).

The identification of functional groups on the surface of biochar was conducted through Fourier-transform infrared (FTIR) spectra analysis using Nicolet 8700 (Nicolet, Madison, WI, USA) operating in potassium bromide tableting mode.

### 2.4. Electrochemical Analysis

The electron exchange capacity of biochar is presently assessed based on its capacity to donate or accept the maximum number of electrons [38]. The electron exchange capacities of the biochar were quantified by mediated electrochemical reduction (MER) and oxidation (MEO), referring to a previously published study [39]. The electrical conductivity (EC) of biochar was determined using the four-probe bed technique suggested by [40].

Cyclic voltammetry (CV) was used to determine the redox potentials in the reactor according to the procedure described by [41]. The samples tested with CV were those reacted for 14 d in anaerobic incubation. CV measurements were conducted under a nitrogen atmosphere at a scan rate of 50 mV $s^{-1}$ in a conventional three-electrode electrochemical cell using a CHI 660C potentiostat (Shanghai CH Instrument Company, Shanghai, China).

### 2.5. Phospholipid Fatty Acid (PLFA) Analysis

Freeze-dried soil samples (2 g) were extracted with chloroform/methanol/citric acid buffer (15 mL at a 1:2:0.8 vol ratio). The extracted fatty acids sequentially eluted with chloroform, acetone and methanol aliquots. Then, the samples were dried under $N_2$. Then, internal standard nonadecanoic acid methyl ester (19:0) was added and used as a reference peak for calculating. A mild alkaline methanol solution converted polar lipids to fatty acid methyl esters (FAMEs). Finally, dried FAMEs were re-dissolved in n-hexane and identified and quantified by an gas chromatograph (Agilent 7890 B, Santa Clara, CA, USA) and MIDI Sherlock Microbial Identification System version 4.5 (MIDI Inc., Newark, DE, USA), respectively. The column was an Agilent 19091B-102E (Agilent, Santa Clara, CA, USA) Ultra 25% Phenyl Methyl Siloxane (25.0 m × 200 μm × 0.33 μm).

### 2.6. Statistical Analysis

The pe + pH value was calculated using the equation of pe + pH = pH + Eh (mV)/59.2. Statistical significance was assessed using Tukey's multiple comparison tests for one-way analysis of variance (ANOVA) or independent-sample $t$ tests ($p < 0.05$). The analyses were performed using SPSS version 16.0 (SPSS Inc., Chicago, IL, USA).

## 3. Results

### 3.1. Changes in $Cd^{2+}$, DOC, $Fe^{2+}$ and $SO_4^{2-}$ Concentrations in Pore Water

The change in the $Cd^{2+}$ concentration of pore water ($W\text{-}Cd^{2+}$) under anaerobic culture for 45 days is presented in Figure 1. The $W\text{-}Cd^{2+}$ showed a rapid decrease in the initial three days. The decline rate of $W\text{-}Cd^{2+}$ in ZZ soil was faster than that in LY. In LY soil, $W\text{-}Cd^{2+}$ decreased by 28.0% on the 3rd day and by 66.2% on the 20th day, and it reached equilibrium on day 30 with an average 75.8% decline. In the ZZ soil, $W\text{-}Cd^{2+}$ decreased sharply by 37.9% and 74.9% on Day 3 and Day 10, respectively, and reached equilibrium on day 20 with an average 85.6% decline. The addition of RSB resulted in a significant decrease in the concentration of $W\text{-}Cd^{2+}$ in both soils. On the first day, RSB application decreased $W\text{-}Cd^{2+}$ by 48.3% and 28.8% in LY and ZZ, respectively, compared to CK. From Day 3 to Day 45, $W\text{-}Cd^{2+}$ in HRSB was lower than that in the LRSB treatment in LY and ZZ soil. On the 10th day, $W\text{-}Cd^{2+}$ in the HRSB treatment was lower by 34.2% on average in soils, compared to LRSB.

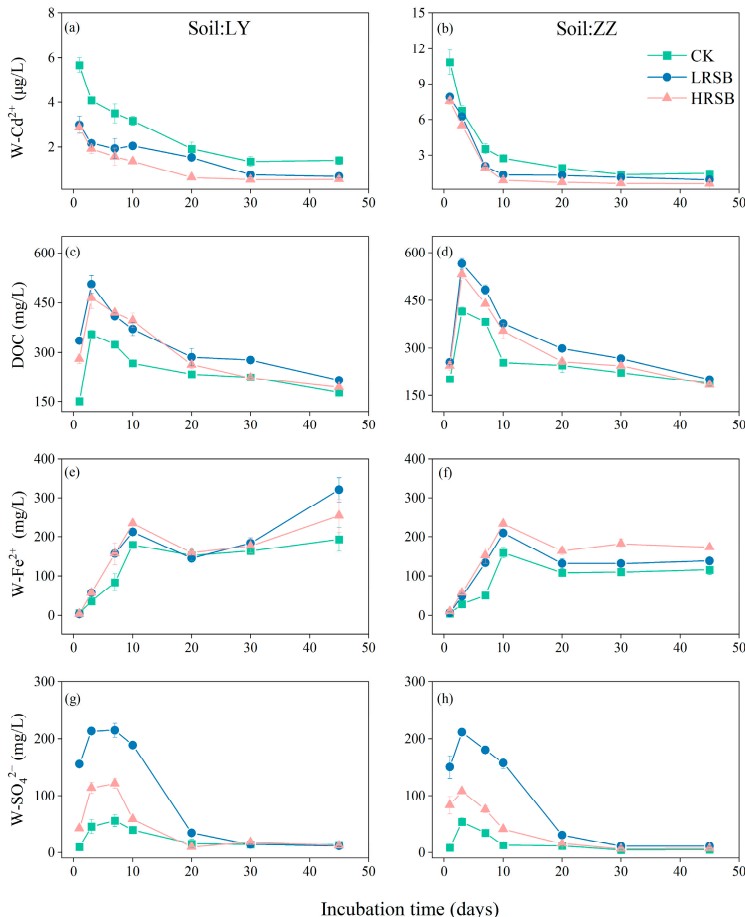

**Figure 1.** Dynamics of W-Cd$^{2+}$ (**a**,**b**), soil DOC (**c**,**d**), W-Fe$^{2+}$ (**e**,**f**), and W-SO$_4{}^{2-}$ (**g**,**h**) in LY and ZZ soils under anoxic conditions for batch experiments. The data are expressed as the mean ± SD, *n* = 3.

The concentration of dissolved organic carbon in soil (DOC) in CK reached the highest value on Day 3 (Figure 1), which in ZZ was higher by 16.6% than that in LY. Compared to CK, DOC in the LRSB and HRSB treatments increased by 39.7% and 30.6% on average in soils, respectively, on Day 3. The concentration of Fe$^{2+}$ in pore water (W-Fe$^{2+}$) increased in the first 10 days, and gradually decreased to reach equilibrium (Figure 1). The concentration of W-Fe$^{2+}$ was significantly increased with the addition of RSB, especially in the HRSB treatment. And, the increase in W-Fe$^{2+}$ in ZZ was higher than that in LY. On the 10th day, the W-Fe$^{2+}$ in the HRSB treatment significantly increased by 30.0% and 46.0% in LY and ZZ, respectively, compared to CK. Meanwhile, the W-Fe$^{2+}$ in the HRSB treatment significantly increased by 10.4% and 10.8% in LY and ZZ, respectively, compared to LRSB. The concentration of SO$_4{}^{2-}$ in soil pore water (W-SO$_4{}^{2-}$) significantly increased during the first 3 days, then gradually decreased to reach equilibrium on the 20th day (Figure 1). On Day 3, compared to CK, the W-SO$_4{}^{2-}$ in the LRSB and HRSB treatments increased by 4.7- and 2.5-fold in LY, and by 3.9- and 2.0-fold in ZZ, respectively.

### 3.2. Changes in pH and Eh in Soil

Eh decreased gradually, while the pH increased during the flooding incubation in Figure 2. The pH increased by 1.0 units in LY and by 1.3 units in ZZ in the first 10 days and then reached equilibrium. RSB addition significantly increased the soil pH. Compared with CK, pH in the biochar treatments increased by 0.51 and 1.0 units on average on the first day in LY and ZZ, respectively. The Eh values in both soils exhibited a significant decrease within the initial 30 days (Figure 2). RSB intensified the decrease in soil Eh, and the decreasing extent in HRSB was higher than that in LRSB. The soil Eh in HRSB reached

equilibrium on Day 10 in ZZ with a 42.8 mV decrease, and on Day 20 in LY with an 89.0 mV decrease, compared to LRSB.

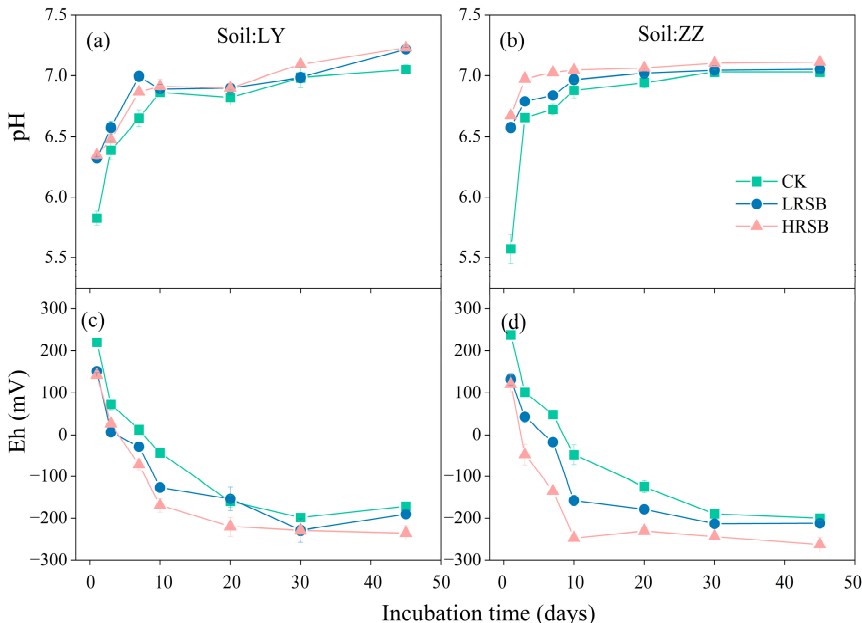

**Figure 2.** Dynamics of pH (**a**,**b**) and Eh (**c**,**d**) in RS and ZZ soils under anoxic conditions for batch experiments. The data are expressed as the mean ± SD, *n* = 3.

### 3.3. Change in the Proportion of Cd Fractions in Flooding Soil

The proportion of the chemical speciation of Cd in soil on Day 45 was determined by the BCR sequential extraction procedure (Figure 3). The proportion of acid-soluble Cd and Fe-Mn oxide Cd accounted for more than 90% of the speciation of Cd in soil. Compared to CK, acid-soluble Cd in the RSB treatments decreased by 58.1% in LY, higher than that in ZZ soils, with a decrease of 31.3% on average. In contrast, the proportion of Cd bound to Fe and Mn oxides in the RSB treatments was higher by 38.5% and 22.6% on average in LY and ZZ soils, respectively, compared to CK.

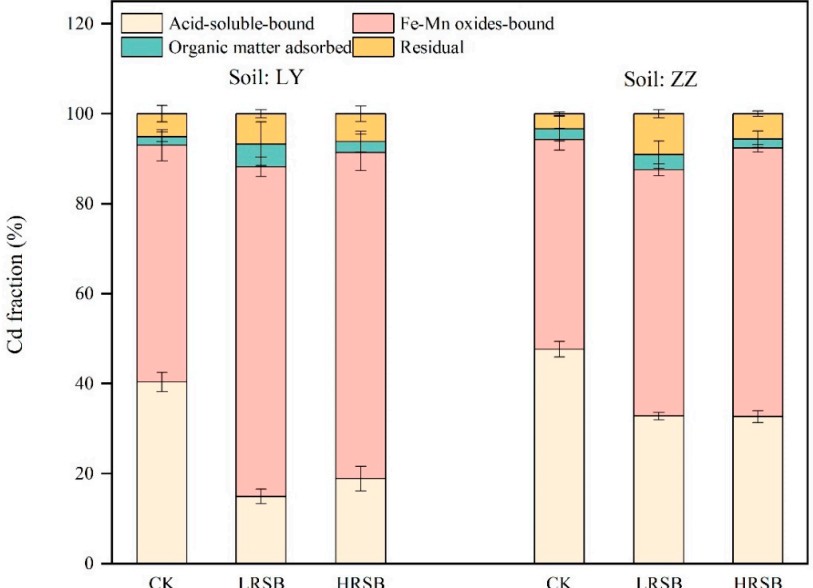

**Figure 3.** The change in Cd fractions under flooding soil for batch experiments. The data are expressed as the mean ± SD, *n* = 3.

### 3.4. Change in the Concentrations of Poorly Crystallized Fe Oxide (Fe$_o$) and Fe-Mn Oxides Cd in Flooding Soil

The Fe$_d$ and Fe$_o$ in LY is higher by 35.3% and 36.6% than that in ZZ (Table 1). RSB addition decreased free Fe oxide (Fe$_d$) concentration but increased Fe$_o$ concentration (Figure 4). The extent of the increase in Fe$_o$ in the RSB treatment was higher in ZZ than that in LY. The LRSB and HRSB addition increased Fe$_o$ by 4.9% and 13.4% in LY, and by 10.5% and 16.8% in ZZ, compared to CK. RSB addition significantly increased the ratio of Fe$_o$/Fe$_d$ in both soils, and the highest increase (38.1%) was found in the HRSB treatment in ZZ.

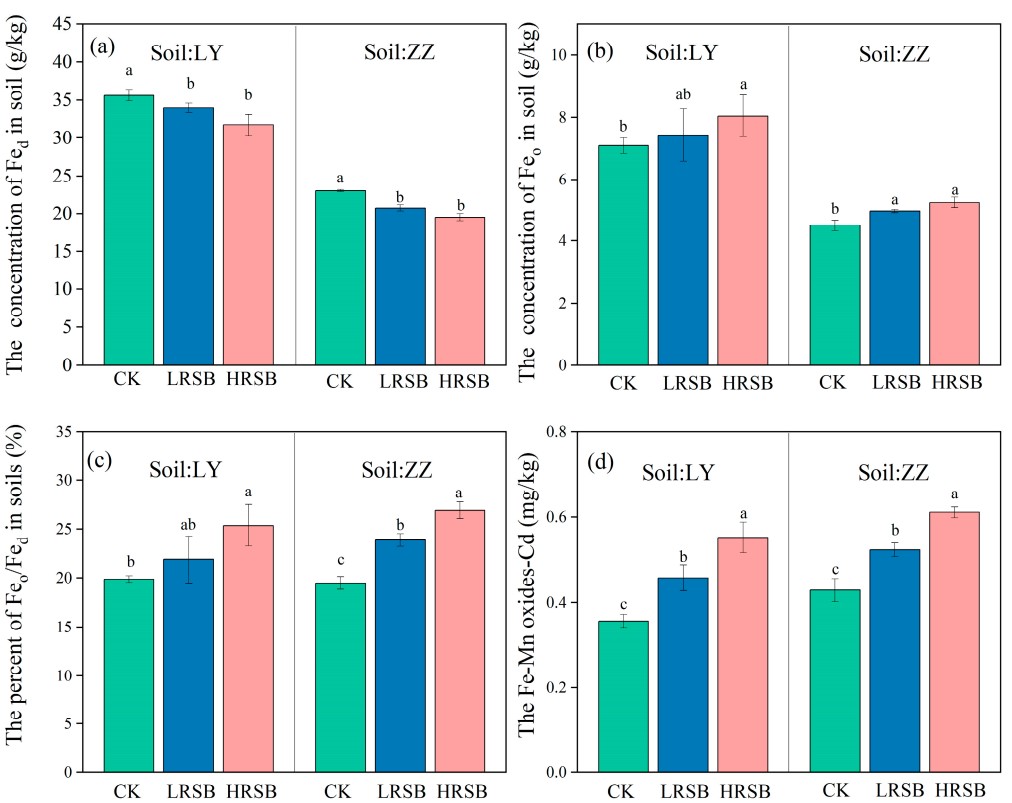

**Figure 4.** Change in the concentration of Fe$_d$ (**a**), Fe$_o$ (**b**), Fe$_o$/Fe$_d$ (**c**) and Fe/Mn-Cd (**d**) under anoxic conditions for batch experiments. The data are expressed as the mean ± SD, *n* = 3. Only significant differences (*p* < 0.05) are shown by the letter (a, ab, b, c) in lowercase on the top of the data column.

The Fe-Mn oxides Cd concentration in the RSB treatments, particularly in the HRSB treatment, increase by 54.6% and by 42.5% in LY and ZZ soils, respectively, compared to CK (Figure 4). The Fe-Mn oxide Cd in the HRSB treatment was higher by 20.6% in LY, and by 16.8% in ZZ soils, respectively, compared to the LRSB treatments.

### 3.5. Concentrations of Acid-Soluble Cd and AVS in Soils

The concentrations of acid-soluble Cd are shown in Figure 5. The concentration of acid-soluble Cd in the LRSB and HRSB treatments significantly decreased by 65.9% and 47.4% in LY, respectively, and by 28.5% and 23.5% in ZZ, respectively (Figure 5). Molybdate addition was found to increase acid-soluble Cd by inhibiting sulfate reduction (Figure 5). In comparison to the biochar treatments with molybdate addition, the acid-soluble Cd concentration in LRSB and HRSB was lower by 37.8% and 27.0% in LY, and by 29.8% and 25.3% in ZZ, respectively. However, acid-soluble Cd in CK was lower by 24.2% and 19.1% in LY and ZZ, respectively, than that in soil with a molybdate addition. This suggested that the contribution of sulfate reduction to Cd immobilization was significantly improved by RSB, especially for LRSB addition in LY soil. The concentrations of available SO$_4^{2-}$ and acid-volatile sulfide (AVS) in soil are shown in Table 1 and Figure 5. LRSB application

increased AVS by 46.4% and 48.9% in LY and ZZ, respectively, compared to CK. AVS in LRSB was 8.4% and 10.8% higher than that of HRSB in LY and ZZ soils, respectively.

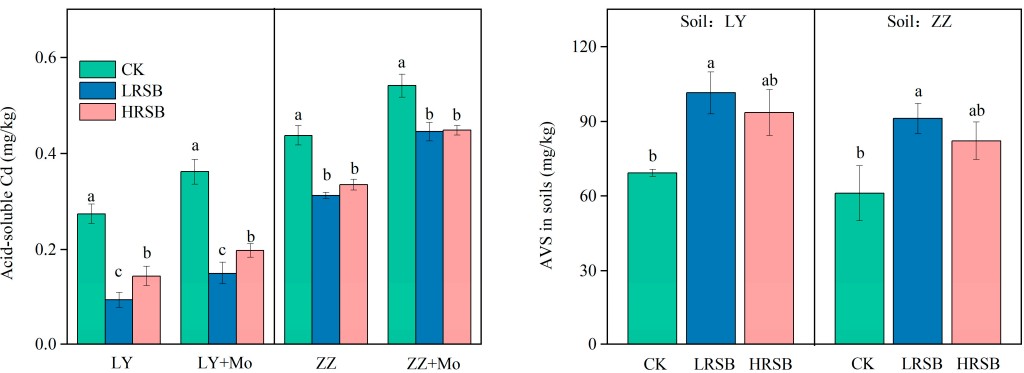

**Figure 5.** The effects of molybdate on the concentration of acid-soluble Cd under flooding soil for batch experiments. The concentration of AVS under flooding soil for batch experiments. The data are expressed as the mean ± SD, *n* = 3. Only significant differences (*p* < 0.05) are shown by the letter (a, ab, b, c) in lowercase on the top of the data column.

*3.6. Change in Soil Microbial PLFAs*

The 20 PLFAs were identified as microbial-based PLFAs and are presented in Figure 6. The total PLFA in ZZ was 2.0-fold higher than that in LY. Compared to CK, total PLFA in the LRSB and HRSB treatments increased by 55.5% and 69.0% in LY and by 36.0% and 12.9% in ZZ, respectively. The PLFA of cy17:0 and cy19:0 in ZZ was 2.3-fold higher than that in LY. The PLFA of cy17:0 and cy19:0 significantly increased by 26.1% and 39.5% in the LRSB and HRSB treatments in LY compared to CK. The PLFA of 18:1$\omega$9 in ZZ was 2.4-fold higher than that in LY. The 18:1$\omega$9 in the LRSB and HRSB treatments increased by 62.4% and 70.4% in LY soil.

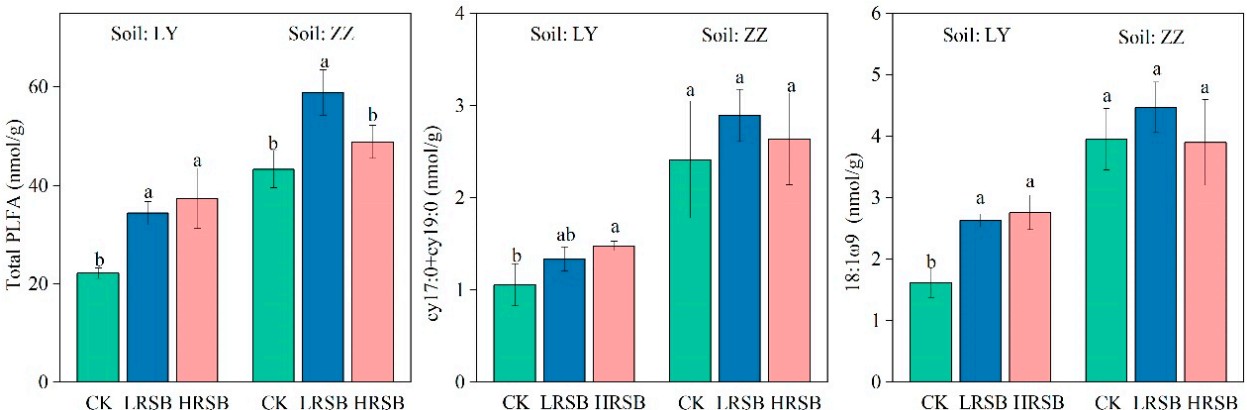

**Figure 6.** The PLFA values of total PLFA, cy17:0 + cy19:0 and 18:1$\omega$9 in soils. The data are expressed as the mean ± SD, *n* = 3. Only significant differences (*p* < 0.05) are shown by the letter (a, ab, b) in lowercase on the top of the data column.

## 4. Discussion

W-Cd$^{2+}$ represents the dynamic changes in soil Cd mobility and species transformation. Our results showed a swift decrease in W-Cd$^{2+}$ in response to the rapid rise in pH and the concurrent decline in Eh (Figures 1 and 2). The consumption of a significant amount of H$^+$ through reducing reactions led to a subsequent increase in pH [42], thereby generating additional sorption sites for Cd binding through ion exchange and cationic complex reactions in soils [2]. Furthermore, the degradation of organic matter provides electrons to electron acceptors like iron oxide and SO$_4^{2-}$, thus raising the levels of Fe$^{2+}$

and $S^{2-}$ in soils and reducing Eh [43]. Reduced sulfate might form CdS [8] or FeS, $FeS_2$ precipitates which could further absorb or co-precipitate with $Cd^{2+}$ in flooding soil [44]. The dissolved $Fe^{2+}$ could form secondary Fe oxides by hydroxylation or precipitating into residual Fe oxides by atom exchange [45,46].

RSB addition intensified the decline in W-$Cd^{2+}$ compared to CK (Figure 2). Biochar possesses the capacity to diminish the mobility of metals through mechanisms that include mineral precipitation, ion exchange, complexation with surface oxygen-containing functional groups, and electron coordination [33,47,48]. The passivation mechanism of biochar on cadmium is affected by pyrolysis temperature. It is reported that the CEC of biochar decreased with increasing pyrolysis temperatures [49]. Similar to our results, the CEC of LRSB was higher than HRSB (Table 2). Biochar application gave rise to soil CEC, intensified the negative surface charge of the soil, and consequently resulted in a reduction in the Cd mobility [50]. Previous research has documented the formation of complexes between Cd with ligands, leading to an enhanced Cd immobility [51]. The dissolved organic carbon in LRSB was higher than that in HRSB (Table 2). Biochar (400–500 °C) exhibited a higher abundance of functional groups compared to biochar derived at other pyrolysis temperatures [52]. These surface functional groups were found to effectively absorb Cd through mechanisms such as electrostatic interaction, ion exchange, and surface complexation [53]. In general, the peak at 3400, 1600, 1100, 1430 $cm^{-1}$ was related to H-bonded OH, esters C=O or aromatic C=C, lignin derivative C-O, aliphatic C-H [21]. The result of FTIR revealed that the intensity of peaks observed in LRSB was greater compared to HRSB (Figure S10). Moreover, RSB addition induced a rapid rise in pH and a decline in Eh; this resulted in a greater reduction in pe + pH (pe + pH = pH + Eh (mV)/59.2) compared to the CK treatment, particularly in the HRSB treatment (Figure 2 and Figure S1). The high pH in the biochar accounted for the sustainable increase in soil pH (Figure 2). In addition, biochar has been found to accept and donate electrons reversibly in many environmentally relevant redox reactions (28). The redox potential within the vicinity of the biochar particle may undergo alterations as organic compound-enriched solutions, comprising cations and anions, permeate through the macro- and meso-pores of the biochar [54]. According to the CV results, RSB with peaks in a more negative direction indicated that it had a more substantial effect on reducing conditions than CK in both soils (Figure S6). It is reported that grass and wood biochar produced at 400−700 °C have extensive quinone/hydroquinone functionalities to transport electrons [40]. Although, HRSB contains very low concentrations of DOC and oxygen-based functional groups. It is reported that the rise in graphitic formations amplifies the conductivity of the biochar (>600 °C) [31]. The result of total electron exchange capacities (EEC) showed that HRSB level was higher than LRSB (Table 2). Furthermore, electrical conductivity (EC) was only detected in HRSB. Our results also showed that the decreasing extent of Eh in HRSB was higher than that in LRSB (Figure 2). Li et al. [55] introduced nitrogen into flooded soil to establish deeper pe + pH conditions, while oxygen was added to soil at 70% field capacity to create higher pe + pH conditions. The Cd availability exhibited a decreasing trend corresponding to the pe + pH levels. We further found a positive correlation between pe + pH and W-$Cd^{2+}$ (LY: $R^2$ = 0.748 $p < 0.01$; ZZ: $R^2$ = 0.860 $p < 0.01$) (Figure S2). The addition of biochar led to a decrease in soil pe + pH, which in turn helped immobilize Cd. In addition, the decrease rate of W-$Cd^{2+}$ in ZZ was higher than that in LY (Figure 1). The pe + pH levels and Cd mobility decreased to lower values in soil with higher organic matter following two-months of flooded cultivation [55]. The total PLFAs was observed to be higher in heavy loam soil than that in sandy loam and medium loam soil [56]. It is rather the slightly higher OC content and the higher loam texture of ZZ that will have caused a larger microbial community (as indicated by the total content of PLFA), resulting in higher microbial (reductive) activity (Figures 1 and 6).

Our results showed that RSB promoted acid soluble Cd transformed to Fe/Mn oxide fraction Cd in flooding soil, particularly in LY soil (Figure 3). A pot experiment inputted $^{112}$Cd into a Cd-contamination soil, which remained submerged before the maturity stage; the newly introduced Cd and legacy Cd transformed from acid-soluble to Fe and Mn oxides

fraction during the tillering to heading stages [57]. The utilization of calcium and sulfur fertilizers can effectively lower the pH level of alkaline soil, but its impact on the availability of cadmium in the soil remains minimal [58]. Prolonged reducing conditions caused a pH increase toward a near-neutral value in acid soils (Figure 2). This provides more sorption sites in the surface of Fe−Mn (oxidro)oxides [2]. Furthermore, the addition of biochar has the potential to influence the immobilization of Cd by altering the Fe speciation. RSB addition significantly increased $Fe_o$ (Figure 4). It has also been reported that Cd adsorption in red paddy soil decreased when removing poorly crystallized Fe oxide [59]. The ratio of $Fe_o/Fe_d$ represents a parameter for evaluating the impact of the redox environment on soil. Our results showed that RSB addition increased the rate of $Fe_o/Fe_d$ (Figure 4). Consistent with our findings, SEM-EDX mapping demonstrated the immobilization of Cd within newly formed magnetite, exhibiting an increased $Fe_o/Fe_d$ ratio under anoxic conditions [15]. The positive correlation between $Fe_o/Fe_d$ and Fe-Mn oxides Cd ($R^2 = 0.913$, $p < 0.01$) indicated that RSB might increase Fe-Mn-Cd through increasing $Fe_o/Fe_d$ (Figure S7 and Figure 4).

RSB addition resulted in a reduction in $Fe_d$ and an elevation in $W-Fe^{2+}$, compared to CK (Figures 1 and 4). The reduction of iron oxides is mainly facilitated by iron-reducing bacteria [60]. The analysis of PLFA revealed a higher abundance of $18:1\omega9$ in ZZ, compared to LY soil. The presence of $18:1\omega9$ is a PLFA marker of Clostridia, which serves as an indicator for iron-reducing bacteria [61]. RSB addition significantly increased $18:1\omega9$ in LY soil (Figure 6). Additionally, it is reported that biochar could influence redox conditions by means of proton-coupled electron exchange [62–64]. Our results also showed that pe + pH was negatively corresponded to $W-Fe^{2+}$ (LY: $R^2 = 0.703$, $p < 0.01$; ZZ: $R^2 = 0.750$, $p < 0.01$), which indicated that biochar could promote formation of $Fe^{2+}$ by enhancing reducing conditions in soils (Figure S5). Dissimilatory Fe(III)-reducing microorganisms possess the ability to transport electrons to electron acceptors ($Fe^{3+}$) as a means to acquire energy for metabolic processes. This electron transfer mechanism could be facilitated by the shuttle effects of biochar [65]. Thus, HRSB had higher redox capacity in transporting electron in microbial reduction, which may promote the reduction. Normally, Fe in oxides is reduced to $Fe^{2+}$ that causes the release of Cd bound on its surface during the flooding incubation [66]. However, our results showed a negative correlation between $W-Fe^{2+}$ and $W-Cd^{2+}$ (LY: $R^2 = 0.530$, $p < 0.01$; ZZ: $R^2 = 0.768$, $p < 0.01$). RSB addition increased $Fe_o$, especially in the HRSB treatments in soils (Figure 4). The poorly crystallized Fe oxide offers more effective adsorption sites for trace metal elements than crystalline iron oxides. This is attributed to its specific surface area and abundant -OH functional groups [67–69]. Released Cd would be incorporated into $Fe_o$ during Fe reduction (Figure S4). The presence of poorly crystallized Fe(II)/Fe(III) mixed minerals, including siderite, vivianite, and goethite, has been observed under anaerobic conditions [11]. These minerals are typically formed through the sorption of dissolved $Fe^{2+}$ onto excess Fe oxide. The increase in percent of Fe(II) in poorly crystalline Fe fractions in the RSB treatment (Figure S8) showed that the content of poorly crystallized Fe(II) miners increased during incubation with flooding. The XRD revealed magnetite $(Fe(II)Fe_2(III)O_4)$ formation during Fe(III) reduction [70]. In addition, poorly crystallized Fe oxide had a higher increase in a flooded soil that had lower initial pH [55]. Poorly crystallized Fe oxide has the ability to adsorb or co-precipitate with soil organic matter, potentially hindering its transformation into crystalline iron oxides [71].

Extended periods of flooding have the potential to reduce Cd mobility in soil through the formation of metal sulfides. The application of gypsum (at a rate of 0.15 and 0.30 g S kg$^{-1}$) decreased Cd mobility by 1.9~7.3% in the rhizosphere [72]. $K_2SO_4$ application (at a rate of 0~0.5 g S kg$^{-1}$) resulted in a significant reduction in available Cd, with a decrease ranging from 27.5–73.8% in both rhizosphere and bulk soil under flooded conditions [73]. LRSB could immobilize Cd by forming more sulfide than HRSB. The $SO_4^{2-}$-S content in LRSB was higher than that in HRSB, and the increase in $W-SO_4^{2-}$ in the LRSB treatment was also higher (Table 1, Figure 1). Moreover, LRSB application increased cy17:0 and cy19:0 in LY soil (Figure 6). Cyclopropyl saturated fatty acids (cy17:0 and cy19:0) were typical for

sulfate-reducing bacteria and anaerobic bacteria [74]. Straw biochar has the ability to boost the activity of sulfate-reducing bacteria in soil [75]. Maize straw biochar also increased the relative abundance of sulfate-reducing bacteria, such as *Desulfatitalea* and *Desulfuromonas*, in soils by providing new carbon sources and new habitat niches [76]. In addition, the redox change (pe + pH) could influence the transformation of sulfate into sulfide, which is a prerequisite for formation of Cd-sulfide in anoxic paddy soils [10,77,78]. Sulfide concentrations exhibited an elevation in response to increasingly reducing conditions [17]. The presence of AVS decreased the availability of metals in the sediment, particularly for divalent metals like Cd, Cu, Ni, and Zn, due to precipitating [79]. AVS in the LRSB treatments was higher compared to that in the HRSB and CK treatment (Figure 5). This showed that LRSB input more sulfate in soils and promoted the formation of sulfide. The molybdate experiment found that the contribution of sulfate reduction to Cd immobilization was significantly improved by RSB, especially in LRSB (Figure 5). There was a negative correlation between concentrations of AVS and acid-soluble Cd in soils ($R^2 = 0.668$, $p < 0.05$) (Figure S9). This suggested that LRSB effectively reduced Cd mobility by introducing a higher amount of sulfur into soil compared to HRSB, thereby promoting the formation of sulfide.

## 5. Conclusions

In anaerobic incubation experiment in Cd-contaminated paddy soils, RSB addition led to a reduction in the acid soluble Cd, accompanied by an increase in the Fe-Mn oxides fractions in soils. However, the passivation mechanism of biochar on the Cd immobilization exhibited variability across different temperatures. The rate of EEC and EC was higher in HRSB compared to LRSB. The CV results also showed that HRSB addition could effectively induce a shift towards lower reducing conditions in the soil. The decrease in pe + pH facilitated Fe oxide reduction in soil. The elevated concentration of $Fe^{2+}$ in the RSB treatments facilitated the formation of $Fe_o$ under neutral conditions. This increased the Cd adsorption of Fe-Mn oxides fraction. The $SO_4^{2-}$-S content in LRSB was higher than HRSB, which had a more pronounced effect on enhancing the contribution of sulfide to the Cd immobility.

Our study showed that the redox capacity of RSB could influence the transfer of electrons and proton in soil, which further enhanced Fe and S transformation and Cd immobilization in flooding conditions. The cultivation of paddy rice is characterized by episodic flooding and drainage phases. The redox capacity of RSB may influence Fe, S and Cd speciation in fluctuated redox conditions. This work highlights the importance of the redox capacity of biochar in affecting the Cd immobilization under flooding conditions, thereby providing new insights into the potential of biochar for Cd remediation in rice cultivation.

**Supplementary Materials:** The following supporting information can be downloaded at: https://www.mdpi.com/article/10.3390/agronomy13112693/s1, Figure S1: Dynamics of pe + pH under anoxic conditions for batch experiments. Figure S2: The correlation analysis of pe + pH and W-$Cd^{2+}$ concentration in LY and ZZ soils. Figure S3: The correlation analysis of DOC and pe + pH in LY and ZZ soils. Figure S4: The correlation analysis of W-$Fe^{2+}$ and W-$Cd^{2+}$ concentration in LY and ZZ soils. Figure S5: The correlation analysis of pe + pH and W-$Fe^{2+}$ concentration in LY and ZZ soils. Figure S6: The CVs in LY and ZZ soils at scan rates of 0.05 V $s^{-1}$. Figure S7: The correlation analysis of percent of $Fe_o/Fe_d$ and Fe/Mn-Cd under anoxic conditions for batch experiments. Figure S8: The percent of $Fe^{2+}$/Total Fe under anoxic conditions for batch experiments. Figure S9: The correlation analysis of AVS and $CaCl_2$-Cd in soil. Figure S10: The FTIR of rape straw biochar.

**Author Contributions:** Formal analysis, R.Y.; funding acquisition, L.L.; investigation, T.S. and C.L.; methodology, Y.W. and L.L.; project administration, L.L.; resources, T.S., Q.L. and C.L.; supervision, G.P.; validation, Y.W. and L.L.; writing—original draft, R.Y.; writing—review and editing, R.B., X.L., X.Z., J.Z., K.C., S.J., Y.W. and L.L. All authors have read and agreed to the published version of the manuscript.

**Funding:** We thank the National Natural Science Foundation of China [grant number 42077148] for funding this research.

**Data Availability Statement:** The datasets used and/or analysed during the current study are available from the corresponding author upon reasonable request.

**Conflicts of Interest:** The authors declare no conflict of interest.

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
