# Peer review of "Rape Straw Biochar Application Enhances Cadmium Immobilization by Promoting Formation of Sulfide and Poorly Crystallized Fe Oxide in Paddy Soils"

_agronomy, doi:10.3390/agronomy13112693_

Round 1
Reviewer 1 Report
This is an interesting and sound study on the diverse impact of biochar on the immobilization of Cd2+ in soil under reducing conditions. With some convincing hypotheses and a good choice of different analytical methods, the authors can present new evidence on the topic.
It is well known that biochar increases the sorption of Cd2+ in soil through the introduction of additional functional groups and exchange capacity as well as the strong impact on soil pH.
However, new and original are further findings that biochar interferes with the redox status and reactions in soil. The authors show that this results in stronger reductive decline of better crystalline Fe oxides and the formation of more amorphous Fe oxides. This, together with the formation of sulfide and sulfide components contribute to a stronger immobilization of Cd2+ in two soils.
The authors present additional findings on chemical soil and biochar properties as well as biological soil properties (PLFA markers) further substantiating their interpretations.
Despite a number of small errors (especially language errors) a few more general things must be revised before the manuscript is ready for publication.
The use of abbreviations is excessive. All must be explained in the text and additionally in the abstract, when used for the first time.
The information in the Material and Methods section is rather limited. Many descriptions of methods lack necessary details. Especially the BCR extraction procedure should be outlined in brief, so that the reader can directly understand how the four fractions were obtained and what they represent. For example, the acid-soluble fraction should be described as the Cd bound to carbonate and cation exchange site, obtained through soil extraction with 0.11 M acetic acid.
The effect of adding molybdate should be briefly outlined in the results section (3.5). Simply referring to a “conventional” and “molybdate” group leaves the reader in uncertainty. What would conventional mean? Conventional agriculture? It needs some time to find out that it is simply samples without molybdate added.
The interpretation of PLFA markers is overused. PLFA do not represent so accurately specific groups of microorganisms. The mentioned cy17:0 and cy19:0 are very general markers for gr-negative bacteria and also anaerobics. However, to say that these PLFAs are indicative of SO42− reduction as electron acceptors in soil is not correct. The increased concentration of these PLFA could possibly point in this direction, but no more than that. The PLFA 18:1w9 (c and t) is known as a marker for fungi but is not an indicator for iron-reducing bacteria. However, the PLFA 18:1w8c indicates methanotrophic bacteria (Börjesson et al. FEMS Microbiol Ecol (2004) 48/3, 305-315). Maybe this was confused.
Detailed comments (line)
Abstract: Explain all abbreviations.
21: The meaning of “Feo/free Fe oxide (Fed) and Fe-Mn Cd” remains unclear.
37: Change to “reduction of iron” , “and Fe3+ are reduced”.
50: Change to “precipitation/reduction/hydroxylation”.
52-77: The authors present several specific information from the literature but immediately give counterexamples that the statement made is not necessarily true. As a reader, however, one is left alone and wonders what use this contradictory information is.
64: Pyrolyzed and not paralyzed.
86: Change to “sieved through a 1 mm sieve”.
101: Revise the incomplete sentence “Each treatment set repetitions.“
117: „Following extraction with a 0.5 M HCl solution.“ This information seems to be duplicated.
118-119: Revise the English language.
122: Better explain the BCR sequential extraction and how W-Cd2+ (= water soluble Cd2+?) was determined.
125: What is „A3“ in the information AAS, A3?
126-127: The description of the analysis with FTIR requires more details.
157: Change to “rapid decrease”.
Fig.1: Here and elsewhere, change the titles on the axes. For example “DOC (mg/L)” and not “The concentration of DOC (mg/L)”.
202: Change to “than that in ZZ”.
203: Change to “In contrast, the proportion of Cd bound to Fe and Mn oxides …”
Fig. 3: Most changes are due to the shift from the acid-soluble Cd to Cd bound to Fe and Mn oxides. Please state in how far this may be due to the effect of increased pH (up to more than one pH unit), possibly lowering the extraction efficiency of acetic acid. Was it clarified in preliminary experiments in how far biochar alone – without the additional effect of reducing conditions – alters the efficiency of the BCR extractants?
224: Change to “are shown”.
226: Rename the “conventional and molybdate groups”.
226 and elsewhere: Correct “was showed in Fig.” to “it is shown in Fig.”.
272: rewrite “in a more decrease”.
272: pe+pH is merged into one number (see respective Figs.). The authors should explain in Materials and Methods how this combined parameter (with very different units) was derived.
281-282: “Higher DOC and total PLFA in ZZ might cause soil pe+pH and W-Cd decreasing faster than that in LY (Figs. 1 and 6).” It is rather the slightly higher OC content and more loamy texture of ZZ that will have caused a larger microbial community (as indicated by the total content of PLFA) resulting in higher microbial (reductive) activity.
286-288: The message doesn’t become clear. Rewrite this section.
297: This information doesn’t match here. Delete. “Wood chips biochar enriched Fe(III)-reducing Geobacteraceae in soil [20]“.
309: Delete “however”.
310: Rewrite this sentence.
314: Change to “which may promote the reduction”.
315: Change to “Fe in oxides is reduced to Fe2+ that causes”
316-317: Give an interpretation for this unexpected correlation.
324: Change to “that the content of poorly crystalline Fe(II) minerals increased in the presence of RSB during incubation with flooding“.
326: Change to “co-precipitate”.
340: Change to “Similarly, straw” (lower case).
350-351: Rewrite this sentence.
360: Replace the comma with a dot.
364-373: Here, the authors present further results that appear as a bonus, without sufficient reference to the preceding discussion. This is particularly regrettable with respect to the FTIR results and should be changed. Revise this whole section.
Conclusions: The Conclusions are insufficient and rather a summarizing list of relevant results. This section needs complete revision.
see questionnaire and comments to authors
Author Response
Reviewer #1:
This is an interesting and sound study on the diverse impact of biochar on the immobilization of Cd2+ in soil under reducing conditions. With some convincing hypotheses and a good choice of different analytical methods, the authors can present new evidence on the topic.
It is well known that biochar increases the sorption of Cd2+ in soil through the introduction of additional functional groups and exchange capacity as well as the strong impact on soil pH.
However, new and original are further findings that biochar interferes with the redox status and reactions in soil. The authors show that this results in stronger reductive decline of better crystalline Fe oxides and the formation of more amorphous Fe oxides. This, together with the formation of sulfide and sulfide components contribute to a stronger immobilization of Cd2+ in two soils.
The authors present additional findings on chemical soil and biochar properties as well as biological soil properties (PLFA markers) further substantiating their interpretations.
Despite a number of small errors (especially language errors) a few more general things must be revised before the manuscript is ready for publication.
The use of abbreviations is excessive. All must be explained in the text and additionally in the abstract, when used for the first time.
Thanks and revised.
The information in the Material and Methods section is rather limited. Many descriptions of methods lack necessary details. Especially the BCR extraction procedure should be outlined in brief, so that the reader can directly understand how the four fractions were obtained and what they represent. For example, the acid-soluble fraction should be described as the Cd bound to carbonate and cation exchange site, obtained through soil extraction with 0.11 M acetic acid.
Thanks and revised (line 120-152). The details were added in Material and Methods.
The effect of adding molybdate should be briefly outlined in the results section (3.5). Simply referring to a “conventional” and “molybdate” group leaves the reader in uncertainty. What would conventional mean? Conventional agriculture? It needs some time to find out that it is simply samples without molybdate added.
Thanks and revised (line 253-262).
The interpretation of PLFA markers is overused. PLFA do not represent so accurately specific groups of microorganisms. The mentioned cy17:0 and cy19:0 are very general markers for gr-negative bacteria and also anaerobics. However, to say that these PLFAs are indicative of SO42− reduction as electron acceptors in soil is not correct. The increased concentration of these PLFA could possibly point in this direction, but no more than that. The PLFA 18:1w9 (c and t) is known as a marker for fungi but is not an indicator for iron-reducing bacteria. However, the PLFA 18:1w8c indicates methanotrophic bacteria (Börjesson et al. FEMS Microbiol Ecol (2004) 48/3, 305-315). Maybe this was confused.
Thanks and SRB and FeRB was removed (line 271-278; 362; 396). The 18:1ω9 is a PLFA marker of Clostridia, which serves as an indicator for iron-reducing bacteria (Akai et al., 2008; Park et al., 2001). In addition, cyclopropyl saturated fatty acids (cy17:0 and cy19:0) were typical for sulfate-reducing bacteria and anaerobic bacteria (Zhou et al., 2009; Fang et al., 2006).
Akai, J., Kanekiyo, A., Hishida, N., Ogawa, M., Naganuma, T., Fukuhara, H., Anawar, H. N. Biogeochemical characterization of bacterial assemblages in relation to release of arsenic from South East Asia (Bangladesh) sediments. Applied Geochemistry. 2008, 23(11), 3177-3186.
Park, H.S., Kim, B.H., Kim, H.S., Kim, H.J., Kim, G.T., Kim, M., Chang, I.S., Yong, K.P., Chang, H.I., 2001. A novel electrochemically active and Fe(III)-reducing bacterium phylogenetically related to clostridium butyricum isolated from a microbial fuel cell. FEMS Microbiol. Lett. 7 (6), 297–306
Zhou, Q.H., He, F., Wang, Y.F., Zhang, L.P., Wu, Z.B. Characteristics of fatty acid methyl esters (FAMEs) and enzymaticactivities in sediments of two eutrophic lakes. Fresenius Environ. Bull. 2009, 18 (7B), 1262–1269.
Fang, J., Chan, O., Joeckel, R.M., Huang, Y., Wang, Y., Bazylinski, D.A., Moorman, T.B., Ang Clement, B.J., 2006. Biomarker analysis of microbial diversity in sediments of a saline groundwater seep of Salt Basin, Nebraska. Organic Geochem. 37 (8), 912–931.
Detailed comments (line)
Abstract: Explain all abbreviations.
Thanks and all abbreviations were explained in abstract.
21: The meaning of “Feo/free Fe oxide (Fed) and Fe-Mn Cd” remains unclear.
Thanks and all abbreviations were explained in abstract. “Feo/free Fe oxide (Fed)” was removed and “Fe-Mn Cd” changed to “Cd bound to Fe and Mn oxides fraction”.
37: Change to “reduction of iron” , “and Fe3+ are reduced”.
Thanks and revised (line 37). The sentence in line 37 change to “Heterotrophic bacteria are primarily responsible for the reductions of iron during flooding conditions, Fe3+ are reduced to Fe2+ by electron transfer in microorganisms”.
50: Change to “precipitation/reduction/hydroxylation”.
Thanks and revised (line 49).
52-77: The authors present several specific information from the literature but immediately give counterexamples that the statement made is not necessarily true. As a reader, however, one is left alone and wonders what use this contradictory information is.
Thanks and revised (line 62; 69-71). The “However” in line 62 changed to “In addition”. The “However” in line 71 was removed. The sentence in line 69-70 was removed.
64: Pyrolyzed and not paralyzed.
Thanks and revised (line 64).
86: Change to “sieved through a 1 mm sieve”.
Thanks and revised (line 86).
101: Revise the incomplete sentence “Each treatment set repetitions.“
Thanks and revised (line 106). The sentence “Each treatment set repetitions” changed to “Each treatment set repetitions three times”.
117: „Following extraction with a 0.5 M HCl solution.“ This information seems to be duplicated.
Thanks and removed (line 127).
118-119: Revise the English language.
Thanks and revised (line 127-130). The sentence changed to “The acid-volatile sulfide (AVS) in the soils was extracted with 10 mL of 6 M HCL. Then the glass reactors were shaken (200 rpm) for 4 h, and solutions were quantified by the methylene blue method”.
122: Better explain the BCR sequential extraction and how W-Cd2+ (= water soluble Cd2+?) was determined.
Thanks and revised (line 131-144; 149). The methods of total Cd, BCR, and W-Cd2+ were added.
125: What is „A3“ in the information AAS, A3?
Thanks and “A3” was removed (line 149).
126-127: The description of the analysis with FTIR requires more details.
Thanks and more details was added (line 151-153).
157: Change to “rapid decrease”.
Thanks and revised (line 185).
Fig.1: Here and elsewhere, change the titles on the axes. For example “DOC (mg/L)” and not “The concentration of DOC (mg/L)”.
Thanks and revised (Fig 1).
202: Change to “than that in ZZ”.
Thanks and revised (line 230).
203: Change to “In contrast, the proportion of Cd bound to Fe and Mn oxides …”
Thanks and revised (line 231).
Fig. 3: Most changes are due to the shift from the acid-soluble Cd to Cd bound to Fe and Mn oxides. Please state in how far this may be due to the effect of increased pH (up to more than one pH unit), possibly lowering the extraction efficiency of acetic acid. Was it clarified in preliminary experiments in how far biochar alone – without the additional effect of reducing conditions – alters the efficiency of the BCR extractants?
Thanks. When soils are flooded, soil pH increases toward a near-neutral value in all treatments. Biochar increased soil pH at the initial period. On Day 45, The soil pH in the biochar treatments exhibits minimal variation compared to that in the CK (LY:0.17 units; ZZ: 0.05 units). We tested proportion of the chemical speciation of Cd with the soil on Day 45, the effect of increased pH may be limited. The soil pH≈7 in all treatments which still suit for acetic acid extraction. In addition, previous flooded batch experiments in our lab showed that, during the day 7 to 30, soil pH in CK and biochar treatments increased by 0.36 and 0.38 units. On Day 7, soil Eh in biochar treatment was lower by 92.2 mV than that in CK. Compared to that on Day 7, concentration of Fe/Mn-Cd on Day 30 increased by 20.6% on average in biochar treatments, and by 9.1% in CK. On Day 30, concentration of acid-soluble Cd in biochar treatment was lower by 10.1% than that in CK (Yuan et al., 2023). Thus, the shift from the acid-soluble Cd to Cd bound to Fe and Mn oxides is credible.
Yuan, R., Si, T., Lu, Q., Bian, R., Wang, Y., Liu, X., Zhang, X., Zheng, J., Cheng, K., Joseph, S., Li, L., Pan, G., 2023. Rape straw biochar enhanced Cd immobilization in flooded paddy soil by promoting Fe and sulfur transformation. Chemosphere.339 139652.
224: Change to “are shown”.
Thanks and revised (line 254).
226: Rename the “conventional and molybdate groups”.
Thanks and revised (line 254-266). The Fig 5 was changed.
226 and elsewhere: Correct “was showed in Fig.” to “it is shown in Fig.”.
Thanks and revised (line 245; 258; 268).
272: rewrite “in a more decrease”.
Thanks and revised (line 314). The sentence “RSB addition induced a rapid rise in pH and a decline in Eh, this resulted in a more decrease in pe+pH than that in the CK treatment” change to “RSB addition induced a rapid rise in pH and a decline in Eh, this resulting in a greater reduction in pe+pH (pe+pH=pH+Eh (mV)/59.2) compared to the CK treatment, particularly in HRSB treatment”.
272: pe+pH is merged into one number (see respective Figs.). The authors should explain in Materials and Methods how this combined parameter (with very different units) was derived.
Thanks and revised (line 178; 314).
281-282: “Higher DOC and total PLFA in ZZ might cause soil pe+pH and W-Cd decreasing faster than that in LY (Figs. 1 and 6).” It is rather the slightly higher OC content and more loamy texture of ZZ that will have caused a larger microbial community (as indicated by the total content of PLFA) resulting in higher microbial (reductive) activity.
Thanks and revised (line 337-341). The sentence “The total PLFAs was observed higher in heavy loam soil than that in sandy loam and medium loam soil (Zhu et al., 2021)” was added to support discussion.
Zhu, Y., Guo, B., Liu, C., Lin, Y., Fu, Q., Li, N., Li, H. Soil fertility, enzyme activity, and microbial community structure diversity among different soil textures under different land use types in coastal saline soil. Journal of Soils and Sediments. 2021, 21(6), 2240-2252.
286-288: The message doesn’t become clear. Rewrite this section.
Thanks and revised (line 348-353). The sentence change to “Prolonged reducing conditions caused pH increased toward a near-neutral value. This provides more sorption sites in the surface of Fe−Mn (oxidro)oxides. Furthermore, addition of biochar has the potential to influence the immobilization of Cd by altering the Fe speciation. RSB addition significantly increased Feo (Fig 4). It has also been reported that Cd adsorption in red paddy soil decreased when removing poorly crystallized Fe oxide”.
297: This information doesn’t match here. Delete. “Wood chips biochar enriched Fe(III)-reducing Geobacteraceae in soil [20]“.
Thanks and removed.
309: Delete “however”.
Thanks and removed (line 326).
310: Rewrite this sentence.
Thanks and revised (line 324-326). The sentence “Poly-condensed aromatic structures produced at higher pyrolysis temperatures of biochar contribute to electron transportation” change to “Although, HRSB contain very low concentrations of DOC and oxygen-based functional groups. It is reported that the rise in graphitic formations amplifies the conductivity of the biochar (>600°C)”.
314: Change to “which may promote the reduction”.
Thanks and revised (line 372). The sentence “HRSB had higher redox capacity in transporting electron in microbial reduction which may promoting reduction of Fe oxides” change to “HRSB had higher redox capacity in transporting electron in microbial reduction which may promote the reduction.”
315: Change to “Fe in oxides is reduced to Fe2+ that causes”
Thanks and revised (line 374).
316-317: Give an interpretation for this unexpected correlation.
Thanks and revised (line 376-380). The sentence “RSB addition increased Feo, especially in the HRSB treatments in soils (Fig. 4). The poorly crystallized Fe oxide offers more effective adsorption sites for trace metal elements than crystalline iron oxides. This is attributed to its specific surface area and abundant -OH functional groups [68-70]. Released Cd would be incorporated into Feo during Fe reduction (Fig S4)” was used to explain the correlation.
324: Change to “that the content of poorly crystalline Fe(II) minerals increased in the presence of RSB during incubation with flooding“.
Thanks and revised (line 384-386).
326: Change to “co-precipitate”.
Thanks and revised (line 388).
340: Change to “Similarly, straw” (lower case).
Thanks and revised (line 399).
350-351: Rewrite this sentence.
Thanks and revised (line 409). The sentence “The molybdate experiment found that LRSB addition induced the higher increase in sulfide contribution to Cd immobilization (Fig 5).” change to “The molybdate experiment found that contribution of sulfate reduction to Cd immobilization was significantly improved by RSB, especially in LRSB.”
360: Replace the comma with a dot.
Thanks and revised (line 301).
364-373: Here, the authors present further results that appear as a bonus, without sufficient reference to the preceding discussion. This is particularly regrettable with respect to the FTIR results and should be changed. Revise this whole section.
Thanks and revised (line 289-333). The passivation mechanism of biochar on cadmium is affected by pyrolysis temperature. The results of properties of biochar were discussed. The FTIR results showed that LRSB contained more functional group than that in HRSB which may absorb more Cd through mechanisms such as electrostatic interaction, ion exchange, and surface complexation (line 309). Also, functional group on the biochar contributed to electron transportation (line 324).
Conclusions: The Conclusions are insufficient and rather a summarizing list of relevant results. This section needs complete revision.
Thanks and revised.
Reviewer 2 Report
The authors present an interesting investigation into the immobilization of cadmium in anaerobic systems with the application of different preparations of rape straw biochar. The study shows clear differences between biochar prepared at low and high pyrolysis temperatures and with a control prepared without added biochar. Differences are observed between different soil types. The authors argue that the immobilization of cadmium is facilitated through a variety of mechanisms, including increased cation exchange capacity in the biochar prepared at lower temperature, the role of biochar in facilitating sulphate reducing bacteria to form suffice with which cadmium for less soluble components, the formation of amorphous iron oxides with more surface area and OH functionality compared to crystalline iron oxides, and a greater sulfur content in rape straw biochar compared to other types of biochar.
The methods are clearly presented and appropriate. The results show clear differences between these treatments and support the conclusions.
Overall, the paper is easy to understand. Some minor language editing is required to improve the paper.
Abstract: Define the LY and ZZ acronyms.
Line 87 "Pyrolyzing" not "Paralyzing"
Line 101: "Each Treatment Set repetitions." is not a complete sentence and I am not sure what is meant.
Line 189. The authors report changes in pH as % changes - pH is a log scale and it is clearer to present these changes in absolute terms indicating the initial and the changed pH values.
Line 237: change "showed" to "presented", or "shown".
Line 284: It is not clear how the discussion of the 110Cd isotope study is connected to the current study.
line 291: It is not clear how the SEM-EDX study is related to the current study.
The English is of high quality but some editing is required to ensure proper usage of tenses.
Author Response
The authors present an interesting investigation into the immobilization of cadmium in anaerobic systems with the application of different preparations of rape straw biochar. The study shows clear differences between biochar prepared at low and high pyrolysis temperatures and with a control prepared without added biochar. Differences are observed between different soil types. The authors argue that the immobilization of cadmium is facilitated through a variety of mechanisms, including increased cation exchange capacity in the biochar prepared at lower temperature, the role of biochar in facilitating sulphate reducing bacteria to form suffice with which cadmium for less soluble components, the formation of amorphous iron oxides with more surface area and OH functionality compared to crystalline iron oxides, and a greater sulfur content in rape straw biochar compared to other types of biochar.
The methods are clearly presented and appropriate. The results show clear differences between these treatments and support the conclusions.
Overall, the paper is easy to understand. Some minor language editing is required to improve the paper.
Abstract: Define the LY and ZZ acronyms.
Thanks and revised (line 16).
Line 87 "Pyrolyzing" not "Paralyzing"
Thanks and revised (line 87).
Line 101: "Each Treatment Set repetitions." is not a complete sentence and I am not sure what is meant.
Thanks and revised (line 106). The sentence “Each treatment set repetitions” changed to “Each treatment set repetitions three times”.
Line 189. The authors report changes in pH as % changes - pH is a log scale and it is clearer to present these changes in absolute terms indicating the initial and the changed pH values.
Thanks and revised (line 214-222). All changes in pH were changed to present in absolute terms.
Line 237: change "showed" to "presented", or "shown".
Thanks and revised (line 264).
Line 284: It is not clear how the discussion of the 110Cd isotope study is connected to the current study.
Thanks and revised (line 343-346). The sentence change to “A pot experiment inputted 112Cd into a Cd-contamination soil which maintained submerge before the maturity stage, the newly introduced Cd and legacy Cd transformed from acid-soluble to Fe and Mn oxides fraction during the tillering to heading stages.” This sentence was used to support the result that “RSB promoted acid soluble Cd transformed to Fe/Mn oxide fraction Cd in flooding soil”.
line 291: It is not clear how the SEM-EDX study is related to the current study.
Thanks and revised (line 356-357). The sentence change to “Consistent with our findings, SEM-EDX mapping demonstrated the immobilization of Cd within newly formed magnetite, exhibiting an increased Feo/Fed ratio under anoxic conditions.” This sentence was used to support the result that RSB addition increased the rate of Feo/Fed, and RSB might increase Fe-Mn-Cd through increasing Feo/Fed.
The English is of high quality but some editing is required to ensure proper usage of tenses.
Thanks and revised.
Reviewer 3 Report
This is an interesting work, although there are certain aspects that the authors could develop.
The abstract is complicated to read and at times difficult to understand, the authors should correct this important section, since it is the one that will decide, to a great extent, the subsequent reading of an paper.
In the materials and methods section, there is a lack of aspects related to the soils used as reference, specifically EC, DOC (Tables 1 and 2). In the case of biochar, SOC and DOC should be expressed in the same units (g kg-1).
In the same section, materials and methods, when describing the anaerobic incubation, the authors do not indicate the diameter of the serum bottle. In itself, 1:1 (w/v) does not necessarily form a 5 cm high layer of water. This will depend on the physical properties of the soil, which are affected when mixing soil with biochar, especially since biochar has a very fine particle size distribution. On the other hand, the authors do not indicate whether they added 2% biochar by weight or volume.
At the end, the effect of increased Cd adsorption with increasing pH is well known, as the authors point out with reference to several studies. The soils taken as reference, LY and ZZ, have acid pH, while the biochar used is alkaline: 8.2 and 9.2 for LRSB and HRSB, respectively. Testing this suspicion with higher pH soils could establish whether the increase in Cd fixation is due to the action of a higher pH or to the action of the biochar (see Li, Y. et al. 2023. The Effects of Calcium and Sulfur Fertilizers Accompanied by Different Side Elements on the Growth and Cd Uptake of Spinacia oleracea Grown in Cd-Contaminated Alkaline Soil. Horticulturae 2023, 9(7), 835. doi.org/10.3390/horticulturae9070835); a possible trial, consisting of adding ZZ and LY CaCO3 to the soils, would test these extremes.
Author Response
This is an interesting work, although there are certain aspects that the authors could develop.
The abstract is complicated to read and at times difficult to understand, the authors should correct this important section, since it is the one that will decide, to a great extent, the subsequent reading of an paper.
Thanks and revised.
In the materials and methods section, there is a lack of aspects related to the soils used as reference, specifically EC, DOC (Tables 1 and 2). In the case of biochar, SOC and DOC should be expressed in the same units (g kg-1).
Thanks and revised (Tables 1 and 2). The basic information of soil CEC and DOC was added, and units of SOC, DOC, and CEC were revised.
In the same section, materials and methods, when describing the anaerobic incubation, the authors do not indicate the diameter of the serum bottle. In itself, 1:1 (w/v) does not necessarily form a 5 cm high layer of water. This will depend on the physical properties of the soil, which are affected when mixing soil with biochar, especially since biochar has a very fine particle size distribution. On the other hand, the authors do not indicate whether they added 2% biochar by weight or volume.
Thanks and revised (line 101-103). The diameter of the serum bottle is 5 cm. The added 2% biochar by weight.
At the end, the effect of increased Cd adsorption with increasing pH is well known, as the authors point out with reference to several studies. The soils taken as reference, LY and ZZ, have acid pH, while the biochar used is alkaline: 8.2 and 9.2 for LRSB and HRSB, respectively. Testing this suspicion with higher pH soils could establish whether the increase in Cd fixation is due to the action of a higher pH or to the action of the biochar (see Li, Y. et al. 2023. The Effects of Calcium and Sulfur Fertilizers Accompanied by Different Side Elements on the Growth and Cd Uptake of Spinacia oleracea Grown in Cd-Contaminated Alkaline Soil. Horticulturae 2023, 9(7), 835. doi.org/10.3390/horticulturae9070835); a possible trial, consisting of adding ZZ and LY CaCO3 to the soils, would test these extremes.
Thanks and revised (line 346). The discussion about impact of pH on Cd immobility in acid and alkaline soil was added in paper.